# RNAthor – fast, accurate normalization, visualization and statistical analysis of RNA probing data resolved by capillary electrophoresis

**Julita Gumna**[1], **Tomasz Zok**[2], **Kacper Figurski**[2], **Katarzyna Pachulska-Wieczorek**[1]*, **Marta Szachniuk**[1,2]*

**1** Institute of Bioorganic Chemistry, Polish Academy of Sciences, Poznan, Poland, **2** Institute of Computing Science, Poznan University of Technology, Poznan, Poland

* kasiapw@ibch.poznan.pl (KPW); mszachniuk@cs.put.poznan.pl (MS)

## Abstract

RNAs adopt specific structures to perform their functions, which are critical to fundamental cellular processes. For decades, these structures have been determined and modeled with strong support from computational methods. Still, the accuracy of the latter ones depends on the availability of experimental data, for example, chemical probing information that can define pseudo-energy constraints for RNA folding algorithms. At the same time, diverse computational tools have been developed to facilitate analysis and visualization of data from RNA structure probing experiments followed by capillary electrophoresis or next-generation sequencing. RNAthor, a new software tool for the fully automated normalization of SHAPE and DMS probing data resolved by capillary electrophoresis, has recently joined this collection. RNAthor automatically identifies unreliable probing data. It normalizes the reactivity information to a uniform scale and uses it in the RNA secondary structure prediction. Our web server also provides tools for fast and easy RNA probing data visualization and statistical analysis that facilitates the comparison of multiple data sets. RNAthor is freely available at http://rnathor.cs.put.poznan.pl/.

**Data Availability Statement:** All relevant data are within the manuscript and its Supporting Information files.

## Introduction

Structural features are of importance for the biological functions of RNA molecules. Specific RNA structures are recognized by RNA binding proteins, ligands, and other RNAs—these interactions impact almost every aspect of cell life or viral replication. Therefore, there is a great interest in developing novel approaches for proper and rapid RNA structure modeling. The computational methods enable the obtaining of good quality models of short RNAs based on sequence only, but the accuracy of structure prediction decreases with the length of RNA molecules [1–5]. The inclusion of RNA structure probing data as pseudo-energy constraints into the thermodynamic folding algorithms significantly improves the accuracy of RNA structure prediction [6, 7]. Among chemical and enzymatic methods, SHAPE (selective 2′-hydroxyl

**Funding:** This work was funded by the National Science Centre Poland (https://ncn.gov.pl/?language=en) in the form of grants awarded to MS (2016/23/B/ST6/03931, 2019/35/B/ST6/03074) and KPW (2016/22/E/NZ3/00426). Funding for the open access charge was provided by the statutory funds of Poznan University of Technology. The funders had no role in study design, data collection and analysis, decision to publish, or preparation of the manuscript.

**Competing interests:** The authors have declared that no competing interests exist.

acylation analyzed by primer extension) [8] and DMS (dimethyl sulfate) mapping [9] are the best validated and most widely used techniques of RNA structure probing *in vitro* and *in vivo* [10, 11]. Besides, the pipelines of SHAPE and DMS probing data incorporation into RNA structure prediction software are well established [6, 12]. DMS modifies the Watson–Crick edge of unpaired adenosines or cytosines, whereas SHAPE reagents create covalent adducts at the 2′-OH group on the RNA sugar ring in a flexibility-sensitive manner [8, 9, 13]. Several SHAPE reagents that differ in their half-life and solubility have been developed until now [14–16]. They act independently from nitrogen base and, consequently, one probing reagent can be used instead of a combination of base-specific chemicals.

Effective detection and quantitative measurement of modification sites are critical for all RNA probing experiments. Typically, RNA chemical modification is followed by a reverse transcription to cDNA that is truncated or mutated at the adducts position [8, 17]. The sites of RT stops in the cDNA can be read-out using the capillary electrophoresis (CE) or next-generation sequencing (NGS) but only the second technique can be used for the detection of adduct-induced mutations [17, 18]. The NGS-based techniques allow genome-wide and transcriptome-wide profiling of RNA structure. The CE is widely used for resolving reactivity data from medium- and low-throughput RNA probing experiments. There are many examples of SHAPE-CE usage for analysis of the structure of many important RNAs, including ribosomal RNAs [19, 20], long noncoding RNAs [21–23], viral RNAs [24–30], and retrotransposon RNAs [31–33]. Besides, CE can also be used for the analysis of RNA probing experiments utilizing other chemical reagents such as CMCT, kethoxal, hydroxyl radicals, and RNases [34–37].

The extraction of quantitative data from CE electropherograms is challenging and requires complicated, multistep analysis of fluorescence signals. Several computational tools can process electropherograms from SHAPE-CE experiments [38–42]. Among them, ShapeFinder [41] and QuShape [42] are the most widely used and yield high-quality SHAPE reactivity data for 300–600 nucleotides in one experiment. Before the incorporation of probing information into the thermodynamic RNA folding algorithms, the reactivity values must be normalized to a uniform scale that is valid for diverse RNAs. Additionally, visual inspection of nonspecific RT strong-stops (non-induced by adduct formation) is required.

Normalization and other quality control steps are very important aspects of structure probing data analysis. Therefore, we developed RNAthor, a user-friendly tool for fast, automatic normalization, and analysis of the CE-based RNA probing data (Fig 1). Features of our tool include (i) normalization of data from several experiments in the box-plot scheme at once, (ii) automatic detection of strong-stops of reverse transcriptase, (iii) reactivity data visualization, (iv) statistical analysis of the results to compare multiple data sets, and (v) RNA secondary structure prediction based on reactivity data.

## Materials and methods

### RNAthor workflow

In the RNAthor workflow, we distinguish five general stages: validation of the input data (ShapeFinder or QuShape file(s) and optionally RNA sequence), exclusion of unreliable data, normalization of probing data, prediction of the secondary structure (optional), and statistical analysis of the normalized data (optional) (Fig 1).

**Validation of the input data.** Initially, the user-uploaded files, resulting from ShapeFinder or QuShape, are parsed, and the basic validation of their format is executed. If, additionally, a sequence is entered, RNAthor checks whether it is RNA and whether it is at least as long as the sequence in the input file(s). A positive validation results in the next step of the

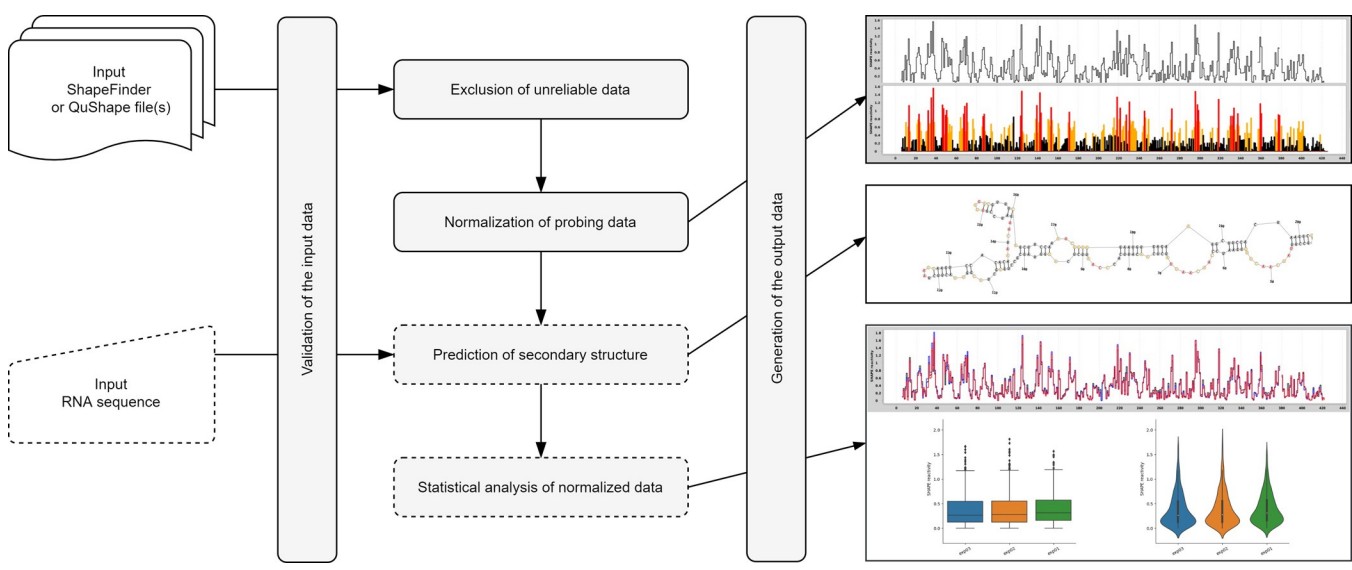

**Fig 1. Workflow in the RNAthor system.**

computational process. Otherwise, the user receives an error message and is asked to provide correct data.

**Exclusion of unreliable data.** Unreliable data usually correspond to premature terminations of primer extension reaction due to reasons other than the formation of the adduct (e.g., preexisting cleavage or modification in RNA). These nucleotide positions are called RT strong-stops. RNAthor offers two ways of detecting such data and excluding them from further processing: a fully automated algorithm and an interactive procedure requiring manual selections. The automated procedure (Fig 2) was implemented based on our experience with analyzing data from RNA chemical probing experiments *in vitro* and *in vivo*. It was optimized for the analysis of SHAPE and DMS probing data. It eliminates the data, which meet one of the following criteria: the absolute reactivity value is negative; the background peak area is at least five times larger than the average background peak area; the difference in peak areas between background and reaction is less than 35% of the average background peak area, and the background peak area in this position is equal to or greater than this average. The alternative is a manual procedure, recommended especially for processing the data from RNA probing experiments other than SHAPE or DMS-probing. In this approach, users can identify unreliable data according to their own experience. They define the negative reactivity threshold, and indicate how to treat the negative reactivity values—they can be left as negative values, changed to 0, or marked as no data. RNAthor displays the histogram with peak areas for modification reaction and background for each nucleotide. Based on this view, users manually select RT strong-stops' positions. All identified RT strong-stops are next excluded from the normalization step.

**Data normalization.** In this step, the data are brought into proportion with one another, and outliers are removed, to provide users with easy to interpret reactivity data on a uniform scale. RNAthor applies the standard box-plot scheme, recommended to normalize the SHAPE-CE data [43, 44]. The normalization process involves: identifying outliers, determining the effective maximum reactivity, and calculating the normalized reactivity values. The initial task is to determine the first (Q1) and the third (Q3) quartile, the interquartile range (IQR), and compute the upper extreme: UP = Q3 + 1.5(IQR). Reactivities greater than UP are considered outliers and not taken into account in subsequent calculations following the principle: for

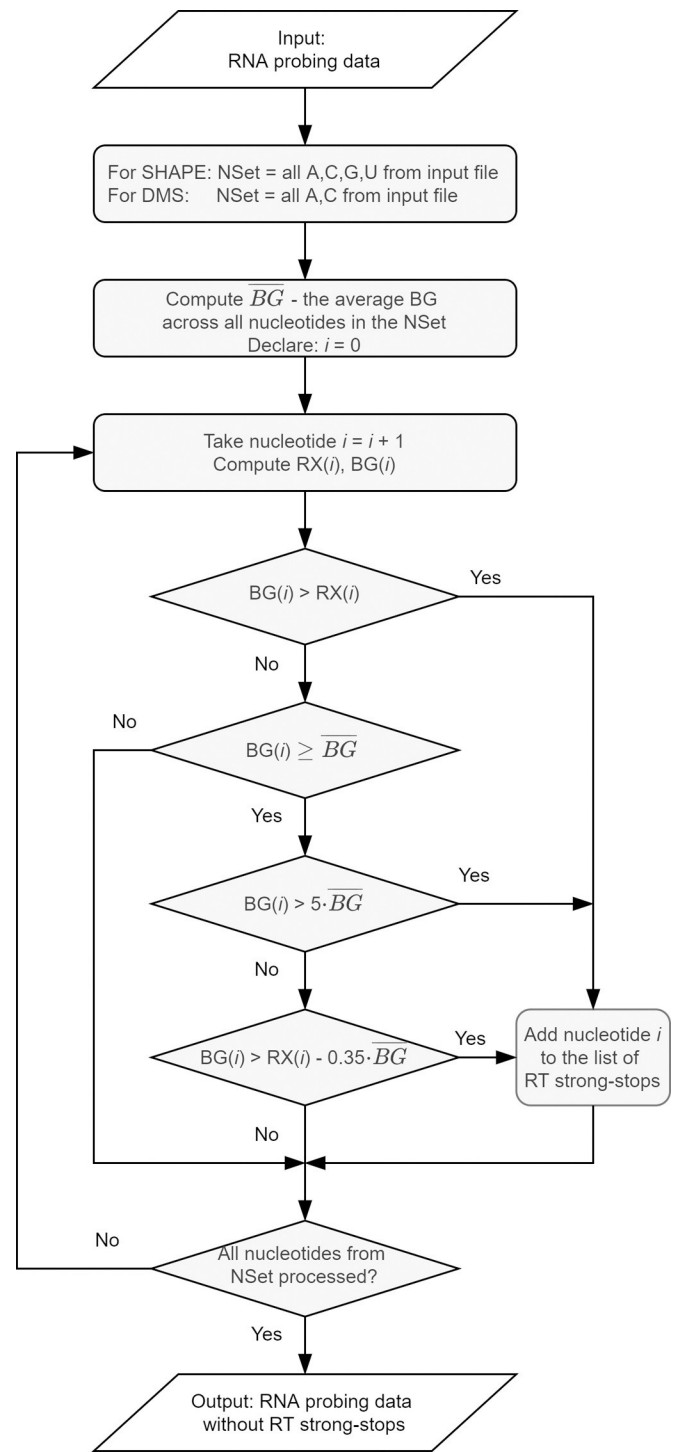

**Fig 2. Scheme of the RNAthor algorithm for unreliable data exclusion.**

RNAs longer than 100 nucleotides, no more than 10% of the data are identified as outliers; for shorter RNAs, maximum 5% of data are removed. The remaining values are used to compute the effective maximum reactivity, i.e., the average of the top 8% of reactivity values. Finally, all absolute reactivity values are divided by the effective maximum reactivity. It results in

obtaining the normalized reactivity data on a uniform scale. Values close to 0 indicate no reactivity (and highly constrained nucleotides), while values greater than 0.85 correspond to high reactivity (and flexible nucleotides).

**Secondary structure prediction.** Optionally, users can obtain the secondary structure predicted for the RNA sequence provided at the input. If the sequence is given, RNAthor automatically executes the incorporated RNAstructure algorithm [45] that supports SHAPE / DMS data-driven prediction. It takes the RNA sequence and the normalized probing data and generates the respective secondary structure. The graphical diagram of the structure is colored according to the color scheme defined for the default reactivity ranges. The output structure is also encoded in the dot-bracket notation.

**Statistical analysis of the normalized data.** Logged users can perform additional statistical analysis of the normalized probing data. The analysis includes 2–5 experiments selected by the user. It consists of running the Shapiro-Wilk test for normal data distribution, Bartlett test of variance homogeneity, non-parametric Mann-Whitney test (if the user selected 2 experiments), and Kruskal-Wallis rank-sum test (if the user selected 3–5 experiments). Two latter tests are performed if the probing data departure from the normal distribution. As a result of the analysis, users receive numerical, textual, and graphical data—among others, the comparative step plot, the box-and-whisker plot, and the violin plot.

## Experimental setup

RNA probing data for the RNAthor validation were obtained from SHAPE-CE and DMS-CE experiments performed in our laboratory for Ty1 RNA (+1–560). The results of SHAPE-based manual analysis were already published in [32]. The DMS experiment was performed especially for this work; its details are presented below. Electropherograms from SHAPE and DMS probing were processed using ShapeFinder software according to the authors' instructions [41].

For RNA probing with DMS, RNA (8 pmol) was refolded in 30 μl of renaturation buffer (10 mM Tris-HCl pH 8.0, 100 mM KCl and 0.1 mM EDTA) by heating for 3 minutes at 95˚C, slow cooling to 4˚C, then adding 90 μl of water and 30 μl of 5x folding buffer (final concentration: 40 mM Tris-HCl pH 8.0, 130 mM KCl, 0.2 mM EDTA, 5 mM $MgCl_2$), followed by incubation for 20 minutes at 37˚C. The RNA sample was divided into two tubes and treated with DMS dissolved in ethanol (+) or ethanol alone (-), and incubated at RT for 1 minute. The reaction was quenched by the addition of 14.7 M β-mercaptoethanol. RNA was recovered by ethanol precipitation and resuspended in 10 μl of water. Primer extension reactions were performed using fluorescently labeled primer [Cy5 (+) and Cy5.5 (-)] as described previously [32]. Sequencing ladders were prepared using primers labeled with WellRed D2 (ddA) and LicorIRD-800 (ddT) and a Thermo Sequenase Cycle Sequencing kit (Applied Biosystems) according to the manufacturer's protocol. Samples were analyzed on a GenomeLab GeXP Analysis System (Beckman-Coulter).

## Web application

RNAthor, implemented as a publicly available web server, has a simple and intuitive interface. It runs on all major web browsers and is accessible at http://rnathor.cs.put.poznan.pl/. The web service is hosted and maintained by the Institute of Computing Science, Poznan University of Technology, Poland.

## Implementation details

The architecture of RNAthor comprises two components: the computational engine (backend layer) and the web application (frontend layer). The backend layer, implemented in Java

OpenJDK 8.0, applies selected modules of the Spring Framework: Spring Boot 2.1.6 enables fast configuration of the application; Spring Security ensures user authentication and basic security; Spring MVC allows compatibility with the Model View Controller and Apache Tomcat server; Spring Test, via Junit and Mockito libraries, enables unit tests and integration; Spring Data allows for comprehensive database services, including transaction management. The user interface (frontend layer) is implemented in Angular technology. User data and basic information about the experiments are collected in the PostgreSQL relational database, the input and output data are saved on the server's hard drive. The tool uses the Apache License 2.0.

## Input and output description

At the input, RNAthor accepts ShapeFinder or QuShape output files in a tab-delimited text format. Users upload their data via the *New experiment* page by selecting 1–15 files from the local folder. All files in the multiple-input should come from several repetitions of the RNA probing experiment performed for the same RNA. Repetitions increase the reliability of structural data for RNA secondary structure prediction. RNAthor processes all the input files in a single run. It starts after data uploading and setting additional parameters for the normalization process (algorithm for RT strong-stops detection, probing reagent, color settings). Additionally, users can provide an RNA sequence that is used to predict the RNA secondary structure.

RNAthor generates a selection of output data. First of all, users obtain the output file in the SHAPE format (*.shape) that is compatible with the RNAstructure software [45]. The file comprises two columns with nucleotide positions and normalized SHAPE / DMS-CE reactivity data. For the multiple-input, the generated SHAPE file contains averaged reactivities from all normalized data. Nucleotides for which there is no reactivity data are assigned -999 values as recommended in [43]. If the user uploaded the sequence of the analyzed RNA molecule, RNAthor provides the RNA secondary structure in dot-bracket notation and the graphical diagram. Additionally, RNAthor generates files that can facilitate the analysis of RNA probing experiments. One of them is the MS Excel file with spreadsheets containing the input data, the normalized reactivity values, and averaged normalized reactivity data with standard deviation (the average and standard deviation are calculated separately for each nucleotide across samples). Each spreadsheet with the input data contains a histogram, identical to this created during manual removal of RT strong-stops. Rows with normalized reactivity values are colored depending on the user's settings. In the processing of large RNAs, this file can help to combine probing data from overlapping reads (with a different set of primers). RNAthor also prepares a graphical output: step plot and bar plot presenting a reactivity profile for one experiment or averaged data from several repetitions. The bar plot is colored depending on the settings: black for reactivities in [0, 0.4), orange for reactivities in [0.4, 0.85), and red for reactivities in [0.85, $\infty$) by default. Logged users that run statistical analysis of experimental data also obtain comparative step plot, box-and-whisker plot, violin plot, and summary of test results. The latter one, available for download in .txt file, informs whether the uploaded data come from a normal distribution, whether they have equal variances, what statistical test was performed, and what is the p-value. The comparative step plot shows reactivity profiles of all compared experiments in one chart. The box-and-whisker plot displays the distribution of data based on the position measures, such as quartiles, minimum, and maximum. The violin plot presents the shape of the distribution and probability density of normalized reactivity values. All generated plots can be saved in PNG or EPS format. Users download the output files separately or in a single zipped archive. They can also obtain them as an email attachment—if the email was provided at the input. Additionally, the email contains a unique link to the result page. The results are

stored in the system for 3 days (for guest users) or 3 months (for logged users). Logged users can extend the storage time by an additional month.

## Results

RNAthor allows for efficient, automated processing and analysis of RNA probing data from SHAPE-CE and DMS-CE experiments and their use in data-driven RNA secondary structure prediction. It was tested on multiple datasets, containing data from SHAPE and DMS probing experiments resolved by capillary electrophoresis. The tests confirmed the reliability of the results and showed the utility of the tool. Here, we describe the experiments performed to compare the results of RNA probing data analysis carried out manually by an expert, and automatically by RNAthor. For the experiments, we chose SHAPE-CE and DMS-CE probing data obtained for RNA of yeast retrotransposon Ty1. The structure of the 5′-end of Ty1 RNA was extensively studied and determined under different experimental conditions and biological states [31, 32, 46].

In the first test, we executed RNAthor for the ShapeFinder-generated files containing the probing data obtained from three independent replicates of SHAPE experiment (raw data used in this experiment are provided in the S1 File). We ran RNAthor with the default settings and the automated algorithm for the identification of RT strong-stops. The generated normalized reactivity data were next compared to the corresponding data published in [32], resulting from manual analysis of the same input. We aligned the obtained bar plots (Fig 3A), and we computed the correlation between normalized reactivity values (Fig 3B). In the second test, we repeated the same procedure for data obtained from the DMS experiment (unpublished data; the experiment was carried out for this work especially). "Blind" human experimentalist analyzed the DMS data preprocessed using ShapeFinder, normalized reactivity values, manually identified unreliable data and applied OriginPro to generate the bar plot presenting the reactivity profile. The results of this manual processing were compared to the output generated by RNAthor that was executed with DMS reagent selected and automated identification of RT strong-stops (Fig 3C and 3D).

From these experiments, we observe that all RT strong-stops identified manually by the expert are also selected for exclusion by the automatic algorithm implemented in RNAthor. On the other hand, few data assigned as RT strong-stops by RNAthor can be considered reliable in the human-dependent analysis. This is due to the rigid criteria for determining RT strong-stops adopted in the algorithm. Table 1 presents the results of the detailed analysis we did by comparing manual, expert-driven, and automatic, RNAthor-performed detection of unreliable data. We computed basic measures used to evaluate the quality of binary classification: true positives (TP)–data classified as reliable by both expert and RNAthor, true negatives (TN)–data classified as unreliable by both expert and RNAthor, false positives (FP)–data indicated as unreliable by the expert but classified as reliable by RNAthor, false negatives (FN)–data indicated as reliable by the expert but classified as unreliable by RNAthor. Using these measures, we calculated the accuracy (ACC), sensitivity (TPR, true positive rate), specificity (TNR, true negative rate), and precision (PPV, positive predictive value) of the automatic algorithm implemented in RNAthor. All these measures were determined for three datasets: SHAPE probing data separately, DMS probing data separately, and data from both sets together. They prove the high quality of the tested algorithm for all datasets. Accuracy and sensitivity equal 0.99, where accuracy, ACC = (TP+TN)/(TP+TN+FP+FN), represents the ratio of correct classifications to the total number of input data, and sensitivity, TPR = TP/(TP+FN), indicates what part of the actual reliable data has been correctly classified by RNAthor. Specificity and precision are both equal to 1, which is because of FP = 0. Specificity, TNR = TN/(TN

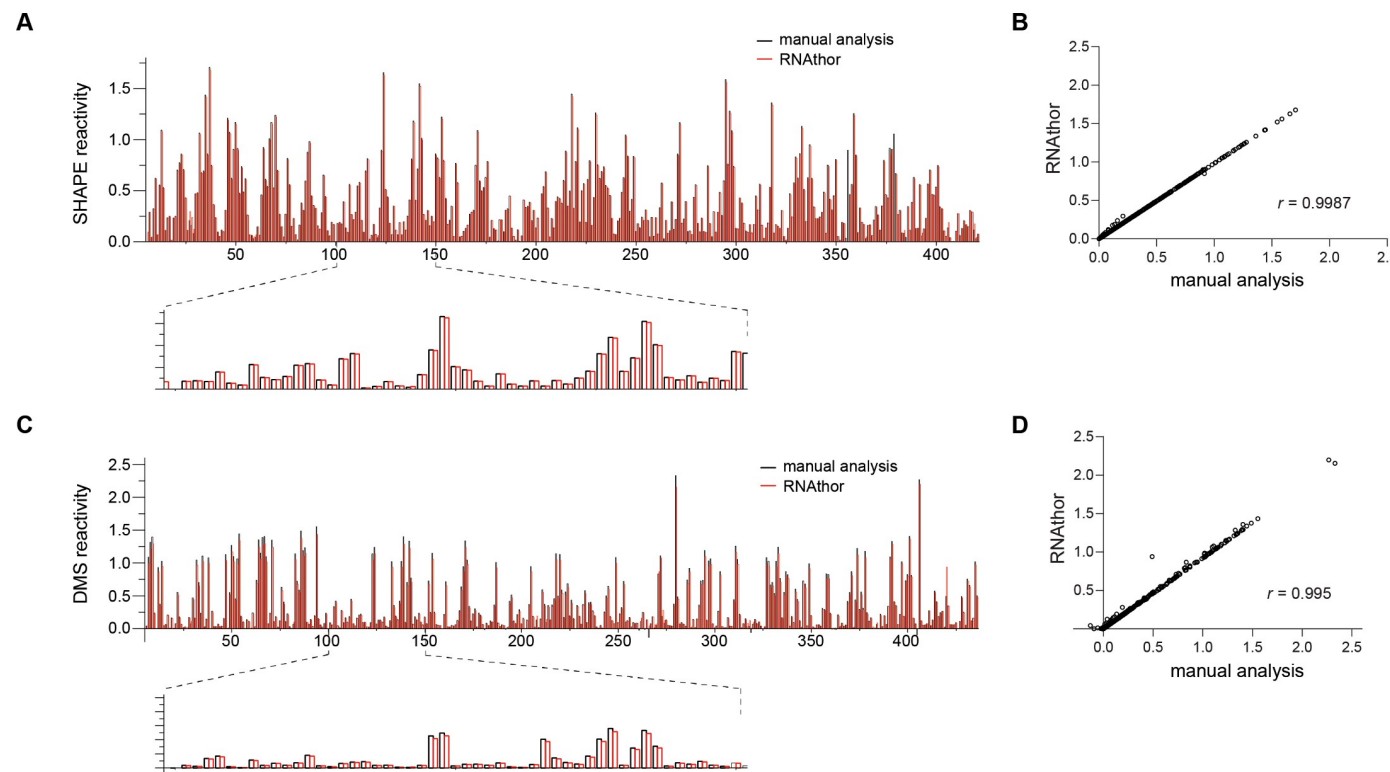

**Fig 3. Automatic and manual normalization of RNA probing data.** SHAPE (A) and DMS (C) reactivity profiles calculated by RNAthor (red) and manually (black). Correlation between RNAthor and manual analysis per nucleotide reactivity estimated for SHAPE (B) and DMS (D).

+FP) is a fraction of correctly classified unreliable data, while the precision, PPV = TP/(TP +FP), informs about the fraction of unreliable data classified as reliable. Finally, the experiments show that—despite some differences between expert- and RNAthor-driven analysis— the normalized RNA probing reactivity values obtained in both approaches are highly similar. The comparison of the reactivity profiles indicates the conformity of manual and automatic procedures. The averaged results from three independent probing experiments yield a Spearman correlation coefficient equal to 0.9987 for SHAPE and 0.995 for DMS-based analysis (Fig 3).

In the testing phase, we also executed statistical analysis to verify the repeatability of obtained results for each nucleotide between the replicates, and compare the reactivity profiles. Fig 4 shows an example of such verification for selected DMS-CE experiments previously performed by RNAthor (raw data used in these experiments are provided in the S1 File). Experiments 1 and 2 (denoted as DMSexp1 and DMSexp2 in Fig 4) were performed under identical experimental conditions, while the higher concentration of DMS was used in experiment 3 (denoted as DMSexp3). We observed a high similarity between reactivity profiles generated for experiments 1 and 2 (Fig 4A), whereas a significant difference was visible for experiment 3

**Table 1. The results of validation of RNAthor algorithm for unreliable data identification.**

| dataset | TP | TN | FP | FN | ACC | PPV | TPR | TNR |
|---------|------|----|----|----|------|-----|------|-----|
| **SHAPE** | 2462 | 38 | 0 | 32 | 0.99 | 1 | 0.99 | 1 |
| **DMS** | 2431 | 35 | 0 | 25 | 0.99 | 1 | 0.99 | 1 |
| **ALL** | 4893 | 73 | 0 | 57 | 0.99 | 1 | 0.99 | 1 |

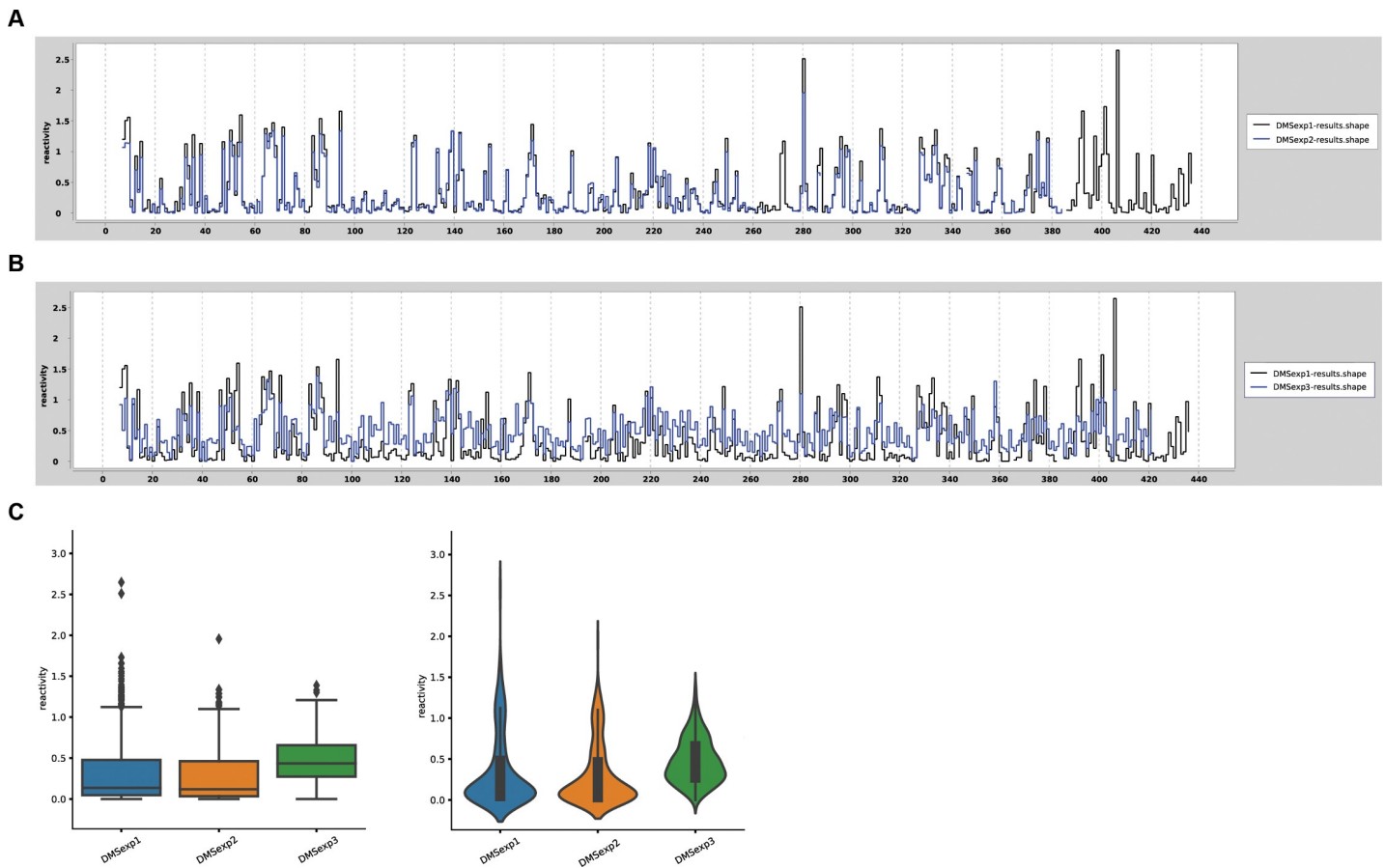

**Fig 4. Example verification of the repeatability of RNA chemical probing experiments.** Comparative step plot for repeatable (A) and non-repeatable (B) replicates of Ty1 RNA probing with DMS. (C) The box-and-whisker plot and violin plot presenting the differences in reactivity data distribution obtained for repeatable and non-repeatable experiments.

(Fig 4B). As expected, the box plot and violin plot present the comparable DMS data distributions for experiments 1 and 2 (Fig 4C). The statistical plots for experiment 3 clearly show a significant increase in the number of more reactive nucleotides, and a concurrent decrease of unreactive nucleotides, consequently, the overall median reactivity is higher (Fig 4C). From these examples, we can see that additional options of RNAthor can be used for fast and easy comparative and statistical analysis for RNA chemical probing experiments.

## Conclusions

In this work, we presented RNAthor, the new computational tool dedicated to the study of RNA structures that enriched the set of web-interfaced bioinformatics systems available within the RNApolis project [47]. RNAthor was designed for a fully automatic, quick normalization, and analysis of SHAPE / DMS-CE data. Although several programs can process the results of CE-based RNA probing, so far, no automatic procedure could identify unreliable data, and this step of the analysis was usually done manually. RNAthor incorporates the algorithm for the automatic exclusion of RT strong-stops to minimize user involvement in the probing data analysis. The tool can be applied to analyze data from other RNA probing methods if capillary electrophoresis and ShapeFinder or QuShape were used for data collection. RNAthor also visualizes the results of RNA probing data normalization, runs the data-driven prediction of

RNA secondary structure, and performs the statistical tests. The latter option facilitates the comparative study of multiple probing experiments, allows to assess the compatibility between experiments, and compare whole data sets of RNAs probed in different experimental conditions (e.g., in vitro, in vivo, ex vivo, in virio, ex virio), or in the absence or presence of protein/ligand. Compared to manual or semi-automated data processing, RNAthor significantly reduces the time needed for data analysis; thus, it can highly improve the study and interpretation of data obtained from RNA chemical probing experiments.

In the future, we plan to extend the functionality of RNAthor by implementing procedures combining RNA probing data from overlapping CE reads to facilitate the structural analysis of large RNAs.

## Supporting information

**S1 File.**
(PDF)

## Author Contributions

**Conceptualization:** Julita Gumna.

**Data curation:** Julita Gumna, Katarzyna Pachulska-Wieczorek.

**Funding acquisition:** Marta Szachniuk.

**Methodology:** Tomasz Zok.

**Project administration:** Katarzyna Pachulska-Wieczorek, Marta Szachniuk.

**Software:** Tomasz Zok, Kacper Figurski.

**Supervision:** Marta Szachniuk.

**Validation:** Julita Gumna, Tomasz Zok, Katarzyna Pachulska-Wieczorek, Marta Szachniuk.

**Visualization:** Kacper Figurski.

**Writing – original draft:** Julita Gumna.

**Writing – review & editing:** Julita Gumna, Katarzyna Pachulska-Wieczorek, Marta Szachniuk.

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
