## [Decision Letter · Decision Letter 0]

9 Jun 2020

PONE-D-20-09557

RNAthor – fast, accurate normalization, visualization and statistical analysis of RNA probing data resolved by capillary electrophoresis

PLOS ONE

Dear Dr. Szachniuk,

Thank you for submitting your manuscript to PLOS ONE. After careful consideration, we feel that it has merit but does not fully meet PLOS ONE’s publication criteria as it currently stands. Therefore, we invite you to submit a revised version of the manuscript that addresses the points raised during the review process.

The revised manuscript should address all the critical points raised by all reviewers.

We look forward to receiving your revised manuscript.

Kind regards,

Danny Barash

Academic Editor

PLOS ONE

Journal Requirements:

Reviewers' comments:

Reviewer's Responses to Questions

**Comments to the Author**

1. Is the manuscript technically sound, and do the data support the conclusions?

Reviewer #1: No

Reviewer #2: Partly

2. Has the statistical analysis been performed appropriately and rigorously? 

Reviewer #1: N/A

Reviewer #2: N/A

3. Have the authors made all data underlying the findings in their manuscript fully available?

Reviewer #1: Yes

Reviewer #2: Yes

4. Is the manuscript presented in an intelligible fashion and written in standard English?

Reviewer #1: Yes

Reviewer #2: Yes

5. Review Comments to the Author

Reviewer #1: Experimental data from RNA structure probing assays in the form of reactivities to structure-sensitive reagents can be integrated with RNA structure-prediction algorithms to improve prediction accuracy. To this end, the raw data is first processed through assay-specific pipelines to get reactivities of nucleotides of RNAs. The reactivities are normalized, and become the input to a structure-prediction algorithm along with the sequences of RNAs of interest. SHAPE-based probing followed by capillary electrophoresis is the traditional way to perform such experiments, and ShapeFinder is a popular tool to estimate reactivities from the electrophoresis data. RNAthor by Gumna et al. primarily serves the purpose of normalizing the reactivities from ShapeFinder and saving files that can be input to RNAstructure. It performs additional visualizations and some statistical tests.

Normalization and exploratory visualization of data are important steps of data analysis. Experimental biologists often struggle with this. Hence, a good interactive web application for this would be a great development. Normalization of such data must account for poor quality information for some nucleotides. Gumna et al. have implemented their empirically developed approach to identify and exclude such nucleotides from analysis. This is the primary technical contribution in this manuscript. However, the approach has not been validated or compared with existing approaches. Tests against some benchmark data and performance assessment are required. Besides, the authors have not described the logic behind their exclusion criteria but simply stated them as mathematical rules. Of the three criteria, the one I could interpret --- filtering negative reactivities --- is common practice in the field. The other two criteria have cutoff values for comparison of peak areas, but it has not been ascertained that they are optimal. They have simply been stated as prior empirical knowledge. Hence, the novelty is low.

Additionally, the authors claim that RNAthor "significantly reduces the time required for data analysis". However, this is not substantiated by any of the results. I'd rather say that an interactive application such as RNAthor could save the time spent writing scripts for data analysis. The current wording would be appropriate if the manuscript presented algorithmic advances that reduce time complexity of the analysis.

Further, the authors have motivated the manuscript in several places by claiming that SHAPE followed by capillary electrophoresis and analyzed by ShapeFinder is a widely recommended approach to study RNA structures. However, the references to support these assertions are often old papers from one particular lab. Hence, the application seems to be of limited interest.

Following are some other comments:

1. It would be nice to have a standalone version of the application.

2. What happens to data uploaded on the server if the user doesn't make an account? Is it also stored on your server for months? Please comment if there any data security issues.

3. Page 10, line 207 says "averaged normalized SHAPE data with standard deviation". Is this referring to average across nucleotides or samples?

4. Page 11, lines 232-239 describe additional statistics computed by RNAthor. However, the purpose behind these specific statistical tests and what the users could do with them is missing.

Reviewer #2: The manuscript by Gumna et al. presents a web-based platform for quality control of RNA structure probing data obtained by experiments that combine SHAPE chemistry with capillary electrophoresis (CE) quantification of the SHAPE reaction’s cDNA products. The platform takes a SHAPE reactivity profile as input and performs automated data normalization, needed to bridge between different experimental conditions and different RNAs, and automated detection of unreliable data points. Users can also visualize the data and run a statistical test that aims to assess reproducibility and possibly also structural variation.

This work tackles a very important aspect of structure probing data analysis, in particular one that has not received sufficient attention to date. Normalization and other quality control steps remain a relatively unexplored area, and researchers often resort to one of a few popular strategies, which many find to be over-simplistic, too narrowly focused, and generally unsatisfactory. As such, this work has the potential to have real impact. However, more work is needed on the authors’ side to bring this work to its full potential, in particular, more testing and a solid convincing demonstration of the utility and validity of the proposed approach.

Several things are missing. First, there are many methods and tools for SHAPE data analysis, currently disregarded by this work. Only 2 software platforms, ShapeFinder and QuShape, are mentioned here, but so many other tools were developed over the last 5-6 years. It is true that newer tools were designed with Seq/MaP/MaPseq protocols in mind; however, this is irrelevant because the proposed platform accepts reactivities as input. It doesn’t matter how reactivates were obtained, as reactivities processed from next-gen or from CE platforms are still reactivities. It has also been shown multiple times (mainly by the Weeks lab) that next-gen-based reactivities have very similar statistical properties to the one obtained by CE platforms. Accounting for other data processing platforms is important because they offer similar normalization routines, and in fact, most of them feature additional popular normalization routines. This, in turn, impacts the novelty of the proposed platform. What I find to be unique to this work is the authors’ approach to automating quality control, particularly the removal of potentially unreliable data points. I don’t think other platforms offer something similar, but this leads me to my second point, which is that the manuscript lacks any demonstration of the performance of the approach (or any other feature unique to this work, such as reproducibility assessment). I understand the authors have gained substantial experience analyzing structure probing data, but the fact that they believe their method “works well” is insufficient for publishing it. The only way to get users to try this out is by showing them, visually and also quantitatively, that the outputs are indeed robust and reliable. I have referred to this issue in more details in my comments below.

Finally, I wonder why the authors limit consideration to both SHAPE data and CE-based platforms. I think the scope of their work could be extended relatively easily. As I mentioned above, any reactivity profile needs to be normalized and quality-controlled. Furthermore, the bigger a dataset is, the more critical automation is, so why not consider the plethora of datasets obtained by high-throughput sequencing-based experiments? I also don’t see a reason to limit this work to SHAPE data. There are many popular alternatives today, including DMS, HRF, and several SHAPE variants, such as NAI. I understand the authors may have tailored their automated QC routine to the special properties of SHAPE, but it is worth testing how well it does on DMS data. Additionally, as shown by several labs, SHAPE-CE, SHAPE-MaP, and SHAPE-Seq all generate very similar data, so why not extend the scope at least in the context of the SHAPE probe?

To summarize, this work needs to be revised to account for a large body of work from labs other than the Weeks lab and for recent advances in structure probing experimental and data analysis capabilities. However, at the same time, it could also leverage the recent expansion in the scope of structure probing to provide a tool of much broader applicability than its current designation, especially because it targets a step in data analysis which has not been adequately addressed to date. Detailed comments and suggestions are below.

There is some novelty in the automation of selection of reactivities for exclusion from analysis based on background signal. To the best of my knowledge, this type of quality control is normally done manually. However, the manuscript and the proposed tool target analysis of relatively short RNAs (no longer than ~300-500 nt) due to SHAPE/DMS and CE limitations. In such cases, manual/visual inspection of the signal and the background traces is not so time-consuming.

The authors also claim that the proposed automation of selection “works well for SHAPE experiments performed in vitro or in vivo”(page 8). There are two key issues with this statement. First, it is not supported by evidence. Second, it is not clear how the authors determine that their method “works well”. I understand that this statement is based on vast experience with SHAPE data analysis, but this is not convincing from a reader’s perspective, and the readers are your potential users. I would like to see numerous real SHAPE data traces (in vitro and in vivo) from more than one lab, for which the automated method works as well as manual correction. This, in turn, requires a suitable quantitative assessment metric. Since currently there is no consensus metric for evaluating “goodness” of SHAPE data, the authors could come up with their own metric, as long as it is appropriate and convincing. Some investigators use SHAPE-directed structure prediction accuracy to benchmark the performance of data processing pipelines. However, I don’t find that convincing (unless differences are truly dramatic) because the NNTM model introduces so much additional complexity and uncertainty to the output. Another way is to show that agreement between replicates improves after the automated routine is applied to the data. Microarray informatics method developers commonly use this approach to demonstrate that a proposed pre-processing step is effective. Existing measures of replicate agreement that are specialized to structure probing data are found in Choudhary et al., Bioinformatics, 2016 and Choudhary et al., Genome Biology, 2019. Alternatively, the authors could propose a novel quantitative measure that captures those data characteristics that make them think it “works well”.

Data normalization: The authors implemented the popular box-plot method, but they caution the reader to avoid using it for RNAs shorter than 300 nt (page 9). So what should a user do if he/she is studying a relatively short RNA? This lack of options severely limits the utility of the proposed platform. Please also see my other comment below regarding normalization strategies that other data analysis platforms offer. How about providing an alternative strategy for relatively short RNAs?

Some statements need to be toned down and/or revised. For example, “Currently, the most common method of RNA chemical probing is SHAPE used in conjunction with capillary electrophoresis (SHAPE-CE)” (Abstract). While it is true that SHAPE used to be the most popular method for a decade or so, DMS appears to be as popular as SHAPE nowadays. I think it is more appropriate to say that one popular reagent choice is SHAPE. Note that there is a similar statement in the Introduction section, which also needs to be reworded. Additionally, I think that over the last 2-3 years, SHAPE/DMS in conjunction with next-gen sequencing (via Seq or MaPseq protocols, and very recently, also via direct RNA nanopore sequencing) has become a much more popular choice than traditional CE-based structure probing. A quick literature search would reveal both the widespread use of DMS for modification and the widespread reliance on NGS for cDNA sequencing. Another statement I found to be somewhat outdated is “ShapeFinder is a popular computational tool for the extraction of quantitative SHAPE reactivity values from raw CE electropherograms.” To the best of my knowledge, many labs use QuShape (a newer and more automated SHAPE-CE analysis software from the Weeks lab) and others use in-house scripts. While ShapeFinder used to be the platform of choice for SHAPE-CE analysis, a quick literature search will show this is no longer the case, especially over the last 2-3 years. Note, however, that I acknowledge there are major issues with QuShape’s performance, as I know that many labs are unhappy with it and seek better alternatives. Finally, I also find the statement “… manual normalization of these values to a uniform scale and exclusion of unreliable data are both required before their usage by RNA structure prediction software” to be somewhat misleading because QuShape does feature reactivity normalization and might also allow users to exclude unreliable data (not sure about the latter, though). The way the abstract is worded, one might think that users currently have no software tools for normalizing the data and possibly also excluding unreliable measurements, and I don’t think that is indeed the case.

In continuation to my previous point, there are multiple software platforms, other than QuShape, which allow users to normalize structure probing data. In contrast to ShapeFinder and QuShape, these platforms were designed for NGS-based probing data, hence the initial input to these tools must take the form of reads, FASTA files, or read counts. However, once reactivities have been calculated, these softwares could be used to apply several different normalizations to them. In other words, reactivity normalization is independent of how reactivities are obtained. They may be obtained from CE or NGS data, using a variety of platforms, but once you have them, they can be normalized by numerous existing platforms. For example, SEQualyzer (Choudhary et al., Bioinformatics, 2016) features 2 normalization strategies: 2-8% and box-plot, RNA Framework (Incarnato et al., NAR, 2018) features 3 normalization strategies: 2-8%, box-plot, and 90% Winsorization, and StructureFold (Tang et al., Bioinformatics, 2015) features (via the Galaxy platform) 2-8% normalization and an option to cap reactivities. Some of these platforms also provide data visualization, for a single experiment and sometimes also for replicates. Some platforms are also open-source (e.g., SEQualyzer and RNA Framework) and this, in turn, allows users to directly use their normalization modules. All this prior work is missing from this manuscript, which might create an impression that users currently have no automated way of normalizing SHAPE/DMS reactivities, which is incorrect. Moreover, the most popular normalization strategies are likely 2-8% and box-plot and are described in detail in Sloma and Mathews, Methods Enzymology, 2015. In fact, one could easily implement them in a Matlab/R/Python script and I disagree with the authors that they “require significant user training and are … time consuming and prone to errors”.

Page 4, second sentence: the description of SHAPE is limited to traditional truncation-based SHAPE chemistry. However, since 2014, modifications are alternatively detected via the MaP approach, where the RT introduces mutations at modified sites. This strategy can only be used in conjunction with DNA sequencing, which is likely why it is not mentioned here. However, for the text to be to be more accurate and up-to-date, I think it should be mentioned.

Results, subsection “A brief overview of SHAPE-CE raw data analysis using ShapeFinder software”: I think this should be omitted or at the least moved to the Supp. Material because knowing how ShapeFinder works is not necessary for understanding the authors’ work. This is because both normalization and detection of unreliable data points (i.e., data QC) occur after the output from ShapeFinder has been obtained. Furthermore, ShapeFinder does not trigger the issues that normalization and data QC address. These issues are inherent to chemical modification experiments.

Results, subsection “RNAthor web application”: I find this material to be suitable for a user manual, especially the description of the various buttons, Reference page, Contact page, and Terms and Conditions page. These are not necessary for understanding the work or getting an idea of how user-friendly it is. It should be possible to compress this subsection into the most essential 4-5 sentences that refer to the ease of use and web interface. I also strongly encourage the authors to compile a short user manual and make it available on the platform’s webpage.

Results, subsection “Input Data”: Similar comments as above. The description is too detailed and should be substantially shortened (e.g., no need to mention which button to press).

Results, subsection “Output Data”: Similar comments as above.

Results, subsection “Output Data”: Note that most or all existing structure probing data analysis pipelines feature data visualization, not only ShapeFinder and QuShape. Some pipelines also allow users to visualize several reactivity profiles together.

Results, subsection “Additional Analysis of Normalized SHAPE-CE Data”: I agree with the authors that SHAPE/DMS data departure from a normal distribution. In fact, it has been rigorously established multiple times that this departure is quite significant (see, for example, Sukosd et al., NAR, 2014, Eddy, Ann. Reviews Biophysics, 2014, Deng et al., RNA, 2016). I therefore do not see why a test for normality is implemented. If the authors have SHAPE data that is nearly Gaussian, I would like to see it appended as Supp. Material.

Results, subsection “Additional Analysis of Normalized SHAPE-CE Data”: To facilitate differential reactivity analysis between two samples and/or to assess reproducibility across experiments, the authors implemented a Mann-Whitney U test, aka Wilcoxon rank-sum test. Note that a very similar test, namely, a Wilcoxon signed-rank test, was recently used for differential reactivity analysis in Choudhary et al., Genome Biology, 2019, albeit applied to similarity scores, not reactivities. The Wilcoxon rank-sum test makes an independence assumption, whereas the signed-rank version of the test relaxes this assumption. Can the authors justify their assumption that the compared samples are independent, especially if these are replicates of the same experiment? Additionally, note that the proposed test does not account for biological variation between samples, since one needs >= 2 replicates from each condition/experiment to assess biological variation per condition. What this means is that if one performs the experiment on two biological replicates (especially relevant to in vivo samples), the test might indicate that results are not reproducible because it picks up the biological variation in the RNA’s structure and its reactivity to the SHAPE reagent. I think it is important to alert users to the limitation of the proposed test and to emphasize what it is capable of detecting. This limitation is inherent to any differential analysis test that compares between reactivity profiles without pre-assessing biological variability between samples in a condition.

As with the automated QC routine, I am missing a demonstration of the utility and the validity of the proposed statistical test. This could be done via benchmarking against existing methods (see, for example, Choudhary et al., Genome Biology, 2019 for a benchmark of recent methods). Potential users need to see some concrete evidence that the proposed test has statistical power. Alternatively, at the least, the authors should show examples of reproducible and irreproducible traces and their corresponding test scores as well as examples of how the test performs on biological replicates vs technical replicates. A comparison with existing reproducibility (QC) measures could also be helpful.

It is clear that the authors have gained tremendous experience analyzing CE-based structure probing data, and that the proposed platform is the result of years of empirical hands-on experience. This is invaluable, especially since there aren’t many labs with that level of expertise. For this reason, this work has the potential to lead to real advances in data analysis. In particular, the authors say “RNAthor was extensively tested on ShapeFinder output files (IPF) from published and unpublished SHAPE-CE experiments performed in our laboratory. It was also tested on IPF files from hydroxyl radicals and DMS probing experiments resolved by capillary electrophoresis.” I think what’s really missing in this manuscript, other than an up-to-date review of existing work, is a demonstration of improved performance on many real datasets from multiple probes, multiple conditions, multiple labs, and multiple RNAs. Such demonstration should also include comparisons to existing alternatives, and there are several ones other than ShapeFinder and QuShape.

6. PLOS authors have the option to publish the peer review history of their article (what does this mean?). If published, this will include your full peer review and any attached files.

Reviewer #1: No

Reviewer #2: No

---

## [Author Response · Author response to Decision Letter 0]

10 Jul 2020

We thank both Referees for a thorough reading of our manuscript “RNAthor – fast, accurate normalization, visualization and statistical analysis of RNA probing data resolved by capillary electrophoresis”. We considered all the remarks, and we substantially revised both - the manuscript and the webserver. The modified fragments of the manuscript are highlighted in red. Below, we respond to all comments of the Referees.

Response to the comments of Referee #1:

Experimental data from RNA structure probing assays in the form of reactivities to structure-sensitive reagents can be integrated with RNA structure-prediction algorithms to improve prediction accuracy. To this end, the raw data is first processed through assay-specific pipelines to get reactivities of nucleotides of RNAs. The reactivities are normalized, and become the input to a structure-prediction algorithm along with the sequences of RNAs of interest. SHAPE-based probing followed by capillary electrophoresis is the traditional way to perform such experiments, and ShapeFinder is a popular tool to estimate reactivities from the electrophoresis data. RNAthor by Gumna et al. primarily serves the purpose of normalizing the reactivities from ShapeFinder and saving files that can be input to RNAstructure. It performs additional visualizations and some statistical tests.

Major comment 1: Normalization and exploratory visualization of data are important steps of data analysis. Experimental biologists often struggle with this. Hence, a good interactive web application for this would be a great development. Normalization of such data must account for poor quality information for some nucleotides. Gumna et al. have implemented their empirically developed approach to identify and exclude such nucleotides from analysis. This is the primary technical contribution in this manuscript. However, the approach has not been validated or compared with existing approaches. Tests against some benchmark data and performance assessment are required. Besides, the authors have not described the logic behind their exclusion criteria but simply stated them as mathematical rules. Of the three criteria, the one I could interpret --- filtering negative reactivities --- is common practice in the field. The other two criteria have cutoff values for comparison of peak areas, but it has not been ascertained that they are optimal. They have simply been stated as prior empirical knowledge. Hence, the novelty is low.

Response: Let us note that no benchmark data from RNA probing are publicly available. Moreover, scientists working with capillary electrophoresis who publish their probing data, share already normalized values. Therefore, we tested RNAthor on “ShapeFinder output files (IPF) from published and unpublished SHAPE-CE experiments performed in our laboratory. It was also tested on IPF files from hydroxyl radicals and DMS probing experiments resolved by capillary electrophoresis.” - this information was already given in the first version of the manuscript. However, we agree that we should have presented the results of our tests in the manuscript. We have filled in this gap now, and, in the Results section, we describe selected tests, present their results and perform a comparative analysis of the results generated by RNAthor with the results from RNA probing data analysis carried out manually by the expert. Note, that such comparison is the only possible one since there are no other automated approaches that combine CE probing data normalization and unreliable data exclusion (this fact was also underlined by the Referee 2). 

Regarding the novelty, let us recall that RNAthor is the first tool providing an automated selection of unspecific RT stops for CE-based RNA probing data. It also gives the possibility of interactive RT stops detection via playing with the automatically generated histogram. Both options are not implemented in any other tool. Therefore, we believe the novelty is undeniable.

Finally, note that many modern computational tools and techniques are based on empirical knowledge or knowledge for which the logic behind is hard to be extracted and hardly understandable by a human. Let us mention the crowdsourcing systems or neural networks. No one challenges these solutions because of the lack of easily explainable logic, as they provide satisfactory solutions to important combinatorial problems. Our method of unreliable data detection resulted from years of experience and long-term decision process. The results of our manual analysis of RNA probing data, based on the rules that are now implemented in the RNAthor algorithm, were published in top scientific journals (e.g., Nature Communications, RNA Biology) and never challenged by experts in the field.

Major comment 2: Additionally, the authors claim that RNAthor "significantly reduces the time required for data analysis". However, this is not substantiated by any of the results. I'd rather say that an interactive application such as RNAthor could save the time spent writing scripts for data analysis. The current wording would be appropriate if the manuscript presented algorithmic advances that reduce time complexity of the analysis.

Response: I suppose that the not very precise wording in the manuscript caused the Referee to misunderstand what we wanted to write here. Therefore, we modified this part of the manuscript. It is much easier and less time consuming to use one program (like RNAthor) that processes the data instead of running three others in sequence and doing the manual analysis of the results. Let us explain how the experimenter, having the ShapeFinder data, should proceed to obtain same results as provided by RNAthor: normalize the absolute SHAPE or DMS reactivities with the spreadsheet, generate the histogram representing peak areas for modification reaction and background for each nucleotide, visually identify unreliable data, prepare the reactivity profiles with a graphical software, run the statistical analysis with a statistical program, prepare a file in the SHAPE data format (*.shape) compatible with the structure prediction software, run RNAstructure with probing data as an additional input. All these tasks take more time when done separately than when they run automatically in one pipeline, like in RNAthor. However, a reliable analysis of time complexity of manual data processing is rather impossible to do.Let us recall that in computing science, time complexity (known also as computational complexity) is estimated based on the number of basic operations (like comparison, simple mathematical operations, elements swapping, etc) done by the algorithm to solve the problem. We can estimate the complexity of all the operations in RNAthor, but we cannot do it for manual data processing. The time spent on manual analysis strongly depends on the experience of the experimenter. In the case of processing data from ShapeFinder, it can take one day for a beginner student and 1-2 hours for a trained experimenter, who has a workshop and scripts ready for simple data processing. Using RNAthor, the same people will do it in 2-3 minutes (if this is not the first time they use the tool) or 30 minutes (if we include the time for reading help).

Major comment 3: Further, the authors have motivated the manuscript in several places by claiming that SHAPE followed by capillary electrophoresis and analyzed by ShapeFinder is a widely recommended approach to study RNA structures. However, the references to support these assertions are often old papers from one particular lab. Hence, the application seems to be of limited interest.

Response: First, let us note that, following good and proper scientific practice, we have cited the original papers, which present the ShapeFinder method - experienced scientists are aware of the fact that this is the right approach (citing the original publication first instead of citing a paper that cites the origin). Obviously, these are papers from one lab where ShapeFinder was developed, and - obviously - the papers are not the recent ones because the method was introduced in 2008. These highly cited examples of ShapeFinder applications are often treated as milestones in RNA probing studies. However, to give examples of SHAPE-CE and ShapeFinder software usage we also cited 4 papers published in 2017 and 2 papers from 2019.

Second, we never claimed that SHAPE followed by capillary electrophoresis and analyzed by ShapeFinder is a widely recommended approach to study RNA structures. In the paper, one could find the following statements: (a) SHAPE-CE with ShapeFinder analysis is popular approach - see citation data below, (b) CE serves as a gold standard method for the single-nucleotide resolution of cDNA fragments from SHAPE or other RNA probing experiments - SHAPE NGS methods were validated using SHAPE-CE, e.g., the accuracy of SHAPEMap with SHAPEMapper1 or SHAPEMapper2 and SHAPE-Seq were tested using capillary electropherogram data (Siegfried et al 2014, Busan et al. 2018, Lucks et al 2011).

ShapeFinder is used in labs that perform capillary electrophoresis and a quick look into the worldwide bibliography, and into the citation record of the original papers proves this fact: the first ShapeFinder paper: 160 citations in Web of Science, 220 in Google Scholar; recent citations dated January 2020. Nevertheless, to solve the Referee’s doubts, we have added the references to the other papers concerning studies performed using ShapeFinder and QuShape. We also added the information on NGS-based RNA probing methods. 

Minor comment 1: It would be nice to have a standalone version of the application.

Response: We have over 10 years of experience in developing bioinformatics programs. During this time, we have developed over 20 different systems for structural bioinformatics: 15 of them are web applications, and 5 are standalone programs. Some reviewers used to ask for the second version of the program: when we publish a web application, they ask for a standalone version, and when we release a standalone version, they require a web application. Such a request, however, should be justified in substance. 

We always try to respond to the needs of our users, and after many years of experience, we know what is critical. A standalone version is very useful when it can be run from a command-line. Such a version is needed when users process huge amounts of data at a time (which is not the case of processing data from capillary electrophoresis) and/or when they do not interfere in the computational process (i.e. do not set a lot of parameters, do not make choices during the calculation process). In the case of interactive programs (RNAthor aims to be interactive), such versions are rather useless. Of course, there are also standalone versions with GUI, not dedicated to high-throughput analysis. They require users to download the program, configure the environment, often also download and install additional software or libraries - many users do not like it. Therefore, in many cases, the best choice is to have a web application. Let us recall the advantages of such solution: all responsibility for the maintenance of the program and the provision of computing power lies with the authors of the program, they can improve the program, remove bugs, add new functions and make the updated version immediately available online (so the users are not requested to update their standalone version, they just have the newest version already on), users do not worry about whether the program is compatible with their operating system and working environment. The only things which are required to use a web application are the internet connection and a web browser - no problem for any scientist these days. We developed RNAthor as the web application taking into account all of these reasons. We do not think that a standalone version is needed at this moment. But in the case of many requests for such a version from our users, we will provide it in the future.

Minor comment 2: What happens to data uploaded on the server if the user doesn't make an account? Is it also stored on your server for months? Please comment if there any data security issues.

Response: The input data and the corresponding results are stored in the RNAthor system for 3 months for registered users (available from their workspace); 3 days for guest users (available on the result page with the unique URL); 24 hours in the case of exemplary input processing (if the user clicks on Run for example SHAPE data or Run for example DMS data; results are available on the result page with the unique URL). Indeed, this information was not complete in the previous version of the RNAthor’s Help file. It is now available in Help section “2.7. Output data”. All the information concerning data issues is also provided in the “Terms and conditions” page (see: http://rnathor.cs.put.poznan.pl/terms). The link to this page is placed in the bottom navigation bar of RNAthor. 

Minor comment 3: Page 10, line 207 says "averaged normalized SHAPE data with standard deviation". Is this referring to average across nucleotides or samples?

Response: The average and standard deviation are calculated separately for each nucleotide across samples. We added this information in the manuscript now.

Minor comment 4: Page 11, lines 232-239 describe additional statistics computed by RNAthor. However, the purpose behind these specific statistical tests and what the users could do with them is missing.

Response: Statistical tests aim to assess reproducibility and possibly also structural variation. Combining statistical information from a probing experiment with the analysis of nucleotide reactivities and the thermodynamic model of RNA folding is also a good and desirable practice. Such an approach can improve the accuracy of probing-directed structure modeling, and allow additional information to be extracted from structure probing data - whether a base is in the single-stranded or double-stranded region. One can find many examples of using similar statistical tests for SHAPE and DMS data analysis e.g. Wilkinson K.A. et al. PLoS Biol. 2008; Purzycka, K.J. et al. Nucleic Acids Res 2013; Li, P. et al. Cell Host Microbe, 2018; Huber, R.G. et al. Nat Commun 2019; Simon, L.M. et al. Nucleic Acids Res 2019 and many others. We reworded the fragment concerning the statistical analysis, and we hope this information is much clearer now.

Response to the comments of Referee #2:

The manuscript by Gumna et al. presents a web-based platform for quality control of RNA structure probing data obtained by experiments that combine SHAPE chemistry with capillary electrophoresis (CE) quantification of the SHAPE reaction’s cDNA products. The platform takes a SHAPE reactivity profile as input and performs automated data normalization, needed to bridge between different experimental conditions and different RNAs, and automated detection of unreliable data points. Users can also visualize the data and run a statistical test that aims to assess reproducibility and possibly also structural variation.

This work tackles a very important aspect of structure probing data analysis, in particular one that has not received sufficient attention to date. Normalization and other quality control steps remain a relatively unexplored area, and researchers often resort to one of a few popular strategies, which many find to be over-simplistic, too narrowly focused, and generally unsatisfactory. As such, this work has the potential to have real impact. However, more work is needed on the authors’ side to bring this work to its full potential, in particular, more testing and a solid convincing demonstration of the utility and validity of the proposed approach.

Comment 1: Several things are missing. First, there are many methods and tools for SHAPE data analysis, currently disregarded by this work. Only 2 software platforms, ShapeFinder and QuShape, are mentioned here, but so many other tools were developed over the last 5-6 years. It is true that newer tools were designed with Seq/MaP/MaPseq protocols in mind; however, this is irrelevant because the proposed platform accepts reactivities as input. It doesn’t matter how reactivates were obtained, as reactivities processed from next-gen or from CE platforms are still reactivities. It has also been shown multiple times (mainly by the Weeks lab) that next-gen-based reactivities have very similar statistical properties to the one obtained by CE platforms. Accounting for other data processing platforms is important because they offer similar normalization routines, and in fact, most of them feature additional popular normalization routines. This, in turn, impacts the novelty of the proposed platform. What I find to be unique to this work is the authors’ approach to automating quality control, particularly the removal of potentially unreliable data points. I don’t think other platforms offer something similar, but this leads me to my second point, which is that the manuscript lacks any demonstration of the performance of the approach (or any other feature unique to this work, such as reproducibility assessment). I understand the authors have gained substantial experience analyzing structure probing data, but the fact that they believe their method “works well” is insufficient for publishing it. The only way to get users to try this out is by showing them, visually and also quantitatively, that the outputs are indeed robust and reliable. I have referred to this issue in more details in my comments below.

Response: First, let us recall that RNAthor has been developed for automatic normalization, visualization and statistical analysis of CE-based RNA probing data, not for NGS-based data analysis. Therefore the tools dedicated for NGS-based probing experiments were not mentioned in the original manuscript. However, we have written about them in the revised version. 

Second, we found the suggestion of the Referee to test the RNAthor normalization function on the NGS data very interesting. Thus, we have investigated the problem but we have found that NGS-platforms have their own normalization tools. Therefore, it is not necessary to implement additional high-throughput data analysis functions in RNAthor - the tool designed for low- and medium-throughput CE-based data processing. Moreover, a comparison of normalization done by RNAthor and NGS platforms is impossible. The platforms for NGS data analysis don’t accept ShapeFinder output files (we tested it for SHAPEMapper and RNA Framework). Interestingly, although the option of the analysis of ShapeFinder files is provided in QuShape it does not work. Summing up, the suggestion that RNAthor should analyse all possible data is out of the question, since even the existing, widely used systems do not do so.

However, we significantly extended the scope of RNAthor by adding new tools that facilitate automatic exclusion of unreliable data and normalization of QuShape output files. We added the possibility to process DMS probing data. Moreover, users can obtain the secondary structure predicted for the RNA sequence provided at the input. RNAthor automatically executes the incorporated RNAstructure algorithm that supports SHAPE / DMS data-driven prediction. The graphical diagram of the structure is colored according to the color scheme defined for the default reactivity ranges. The output structure is also encoded in the dot-bracket notation.

Third, we tested RNAthor on “ShapeFinder output files (IPF) from published and unpublished SHAPE-CE experiments performed in our laboratory. It was also tested on IPF files from hydroxyl radicals and DMS probing experiments resolved by capillary electrophoresis.” - this information was already given in the first version of the manuscript. However, we agree that we should have presented the results of our tests in the manuscript. We have filled in this gap now, and, in the Results section, we describe selected tests, present their results and perform a comparative analysis of the results generated by RNAthor with the results from RNA probing data analysis carried out manually by the expert. Note, that such comparison is the only possible one since there are no other automated approaches that combine CE probing data normalization and unreliable data exclusion.

Comment 2: Finally, I wonder why the authors limit consideration to both SHAPE data and CE-based platforms. I think the scope of their work could be extended relatively easily. As I mentioned above, any reactivity profile needs to be normalized and quality-controlled. Furthermore, the bigger a dataset is, the more critical automation is, so why not consider the plethora of datasets obtained by high-throughput sequencing-based experiments? I also don’t see a reason to limit this work to SHAPE data. There are many popular alternatives today, including DMS, HRF, and several SHAPE variants, such as NAI. I understand the authors may have tailored their automated QC routine to the special properties of SHAPE, but it is worth testing how well it does on DMS data. Additionally, as shown by several labs, SHAPE-CE, SHAPE-MaP, and SHAPE-Seq all generate very similar data, so why not extend the scope at least in the context of the SHAPE probe?

Response: RNAthor was never limited to the SHAPE data. In the first version of the manuscript, we have already written that RNAthor was extensively tested using SHAPE, DMS and hydroxyl radical data. To make it clear now, we have added the DMS example in the webserver, so that the user can try the tool for these data as well. We have also added an option to enable automatic selection of DMS reactivities for an exclusion. See also our response to Comment 1, concerning NGS data processing.

Comment 3: To summarize, this work needs to be revised to account for a large body of work from labs other than the Weeks lab and for recent advances in structure probing experimental and data analysis capabilities. However, at the same time, it could also leverage the recent expansion in the scope of structure probing to provide a tool of much broader applicability than its current designation, especially because it targets a step in data analysis which has not been adequately addressed to date. Detailed comments and suggestions are below.

Response: Please see our response to comment 1 and 2.

Comment 4: There is some novelty in the automation of selection of reactivities for exclusion from analysis based on background signal. To the best of my knowledge, this type of quality control is normally done manually. However, the manuscript and the proposed tool target analysis of relatively short RNAs (no longer than ~300-500 nt) due to SHAPE/DMS and CE limitations. In such cases, manual/visual inspection of the signal and the background traces is not so time-consuming.

Response: Let us explain that ~300-500 nt is the length of one CE read, not the length of analyzed RNA. Users can process much longer RNAs; using normalized overlapping CE reads we analyze RNAs > 5kb nucleotide-long (Andrzejewska et al., in revision). RNAthor is not limited to short RNAs - we did not write anything like this in the manuscript. Note also that RNAthor offers more than just an automated selection of reactivities. We recommend to look at the charts, violin plots, or box plots prepared by our tool and try the statistical analysis. These are very useful features of RNAthor. We hardly believe anyone does all of this manually, even for relatively short RNAs.

Comment 5: The authors also claim that the proposed automation of selection “works well for SHAPE experiments performed in vitro or in vivo”(page 8). There are two key issues with this statement. First, it is not supported by evidence. Second, it is not clear how the authors determine that their method “works well”. I understand that this statement is based on vast experience with SHAPE data analysis, but this is not convincing from a reader’s perspective, and the readers are your potential users. I would like to see numerous real SHAPE data traces (in vitro and in vivo) from more than one lab, for which the automated method works as well as manual correction. This, in turn, requires a suitable quantitative assessment metric. Since currently there is no consensus metric for evaluating “goodness” of SHAPE data, the authors could come up with their own metric, as long as it is appropriate and convincing. Some investigators use SHAPE-directed structure prediction accuracy to benchmark the performance of data processing pipelines. However, I don’t find that convincing (unless differences are truly dramatic) because the NNTM model introduces so much additional complexity and uncertainty to the output. Another way is to show that agreement between replicates improves after the automated routine is applied to the data. Microarray informatics method developers commonly use this approach to demonstrate that a proposed pre-processing step is effective. Existing measures of replicate agreement that are specialized to structure probing data are found in Choudhary et al., Bioinformatics, 2016 and Choudhary et al., Genome Biology, 2019. Alternatively, the authors could propose a novel quantitative measure that captures those data characteristics that make them think it “works well”.

Response: First, let us note that in this comment the Referee denies what he/she wrote in Comment 1. Let us quote from Comment 1 “It doesn’t matter how reactivates were obtained…”, “reactivity normalization is independent of how reactivities are obtained”. According to this, reactivities obtained in vitro, in vivo, ex vivo, in virio, ex virio are still reactivities and do not require separate routines of normalization or selection of unreliable data. 

Second, we agree that we should have provided the results of RNAthor testing. Therefore, we have revised the Results section and we included the results of computational tests performed with RNAthor to validate its reliability. Let us add that no benchmark data from RNA probing are publicly available. Generally, if the authors provide (as supplementary data) reactivity values they are already normalized and unreliable reactivities are set to -999. Therefore, wide computational tests of our tool were hardly possible based on what we found in the worldwide repositories of RNA probing data. To validate the automatic algorithm of RNAthor, we used in vitro and in vivo SHAPE data traces from our lab, which come from real experiments and are of very good quality. Researchers from our labs have a solid background and years of experience in RNA structure probing using SHAPE-CE, DMS-CE and HR-CE. We checked the validation of other tools for RNA probing data analysis (ShapeMapper1 and 2, ShapeFinder, QuShape, RNA Framework) and we found that these tools were also tested using diverse RNAs probed only in the labs of the software’s authors. 

Finally, the approach discussed by the Referee, measuring the replicate agreement (SEQualyzer), serves for quality control and exploratory analysis of high-throughput RNA structural profiling data, whereas RNAthor was designed for low-throughput data analysis and visualization. Therefore, we do not find it useful in our case. 

Comment 6: Data normalization: The authors implemented the popular box-plot method, but they caution the reader to avoid using it for RNAs shorter than 300 nt (page 9). So what should a user do if he/she is studying a relatively short RNA? This lack of options severely limits the utility of the proposed platform. Please also see my other comment below regarding normalization strategies that other data analysis platforms offer. How about providing an alternative strategy for relatively short RNAs?

Response: We thank the Referee for this comment. We improved the normalization method in RNAthor to allow normalization of short RNAs as well. According to a common approach in the box-plot method of normalization, outliers are determined as follows: for RNAs longer than 100 nucleotides no more than 10% of the data are removed, while for shorter RNAs maximum 5% of data are removed. We added this information in the manuscript.

Comment 7: Some statements need to be toned down and/or revised. For example, “Currently, the most common method of RNA chemical probing is SHAPE used in conjunction with capillary electrophoresis (SHAPE-CE)” (Abstract). While it is true that SHAPE used to be the most popular method for a decade or so, DMS appears to be as popular as SHAPE nowadays. I think it is more appropriate to say that one popular reagent choice is SHAPE. Note that there is a similar statement in the Introduction section, which also needs to be reworded. Additionally, I think that over the last 2-3 years, SHAPE/DMS in conjunction with next-gen sequencing (via Seq or MaPseq protocols, and very recently, also via direct RNA nanopore sequencing) has become a much more popular choice than traditional CE-based structure probing. A quick literature search would reveal both the widespread use of DMS for modification and the widespread reliance on NGS for cDNA sequencing. Another statement I found to be somewhat outdated is “ShapeFinder is a popular computational tool for the extraction of quantitative SHAPE reactivity values from raw CE electropherograms.” To the best of my knowledge, many labs use QuShape (a newer and more automated SHAPE-CE analysis software from the Weeks lab) and others use in-house scripts. While ShapeFinder used to be the platform of choice for SHAPE-CE analysis, a quick literature search will show this is no longer the case, especially over the last 2-3 years. Note, however, that I acknowledge there are major issues with QuShape’s performance, as I know that many labs are unhappy with it and seek better alternatives. Finally, I also find the statement “… manual normalization of these values to a uniform scale and exclusion of unreliable data are both required before their usage by RNA structure prediction software” to be somewhat misleading because QuShape does feature reactivity normalization and might also allow users to exclude unreliable data (not sure about the latter, though). The way the abstract is worded, one might think that users currently have no software tools for normalizing the data and possibly also excluding unreliable measurements, and I don’t think that is indeed the case.

Response: Following the Referee’s suggestion, we have revised many statements in the manuscript. Moreover, we have substantially extended the functionality of RNAthor and - currently - it can also process the DMS data and SHAPE data obtained from QuShape. We find ShapeFinder a popular tool, but it is true that QuShape also has many supporters. Let's take a look at the facts: according to the Web of Science, ShapeFinder paper has 160 citations (recent ones from January 2020), QuShape paper - 109 citations. As for the in-house scripts, it is rather hard to discuss their popularity, since such solutions are used only locally. 

Comment 8: In continuation to my previous point, there are multiple software platforms, other than QuShape, which allow users to normalize structure probing data. In contrast to ShapeFinder and QuShape, these platforms were designed for NGS-based probing data, hence the initial input to these tools must take the form of reads, FASTA files, or read counts. However, once reactivities have been calculated, these softwares could be used to apply several different normalizations to them. In other words, reactivity normalization is independent of how reactivities are obtained. They may be obtained from CE or NGS data, using a variety of platforms, but once you have them, they can be normalized by numerous existing platforms. For example, SEQualyzer (Choudhary et al., Bioinformatics, 2016) features 2 normalization strategies: 2-8% and box-plot, RNA Framework (Incarnato et al., NAR, 2018) features 3 normalization strategies: 2-8%, box-plot, and 90% Winsorization, and StructureFold (Tang et al., Bioinformatics, 2015) features (via the Galaxy platform) 2-8% normalization and an option to cap reactivities. Some of these platforms also provide data visualization, for a single experiment and sometimes also for replicates. Some platforms are also open-source (e.g., SEQualyzer and RNA Framework) and this, in turn, allows users to directly use their normalization modules. All this prior work is missing from this manuscript, which might create an impression that users currently have no automated way of normalizing SHAPE/DMS reactivities, which is incorrect. Moreover, the most popular normalization strategies are likely 2-8% and box-plot and are described in detail in Sloma and Mathews, Methods Enzymology, 2015. In fact, one could easily implement them in a Matlab/R/Python script and I disagree with the authors that they “require significant user training and are … time consuming and prone to errors”.

Response: First, let us note that in this comment the Referee denies what he/she wrote in the beginning of the second paragraph of THIS review: “This work tackles a very important aspect of structure probing data analysis, in particular one that has not received sufficient attention to date. Normalization and other quality control steps remain a relatively unexplored area, and researchers often resort to one of a few popular strategies, which many find to be over-simplistic, too narrowly focused, and generally unsatisfactory.” Second, in our opinion an analysis of NGS-based data is outside the scope of this work. See also our response to comments 1, 2 and 5. 

Comment 9: Page 4, second sentence: the description of SHAPE is limited to traditional truncation-based SHAPE chemistry. However, since 2014, modifications are alternatively detected via the MaP approach, where the RT introduces mutations at modified sites. This strategy can only be used in conjunction with DNA sequencing, which is likely why it is not mentioned here. However, for the text to be to be more accurate and up-to-date, I think it should be mentioned.

Response: We thank the Referee for this comment. We introduced additional descriptions in the manuscript.

Comment 10: Results, subsection “A brief overview of SHAPE-CE raw data analysis using ShapeFinder software”: I think this should be omitted or at the least moved to the Supp. Material because knowing how ShapeFinder works is not necessary for understanding the authors’ work. This is because both normalization and detection of unreliable data points (i.e., data QC) occur after the output from ShapeFinder has been obtained. Furthermore, ShapeFinder does not trigger the issues that normalization and data QC address. These issues are inherent to chemical modification experiments.

Response: Indeed, the Results section was not the right place to describe the analysis of SHAPE-CE data with ShapeFinder. We substantially revised the manuscript and this part was removed.

Comment 11: Results, subsection “RNAthor web application”: I find this material to be suitable for a user manual, especially the description of the various buttons, Reference page, Contact page, and Terms and Conditions page. These are not necessary for understanding the work or getting an idea of how user-friendly it is. It should be possible to compress this subsection into the most essential 4-5 sentences that refer to the ease of use and web interface. I also strongly encourage the authors to compile a short user manual and make it available on the platform’s webpage.

Results, subsection “Input Data”: Similar comments as above. The description is too detailed and should be substantially shortened (e.g., no need to mention which button to press).

Results, subsection “Output Data”: Similar comments as above.

Results, subsection “Output Data”: Note that most or all existing structure probing data analysis pipelines feature data visualization, not only ShapeFinder and QuShape. Some pipelines also allow users to visualize several reactivity profiles together.

Response: According to the Referee’s suggestion, we have removed the detailed description of the web application from the manuscript. Also the fragments about input and output data were reduced. Some fragments were moved to Help, which already existed before, but many details were missing there. Therefore, the RNAthor Help was also substantially extended.

Comment 12: Results, subsection “Additional Analysis of Normalized SHAPE-CE Data”: I agree with the authors that SHAPE/DMS data departure from a normal distribution. In fact, it has been rigorously established multiple times that this departure is quite significant (see, for example, Sukosd et al., NAR, 2014, Eddy, Ann. Reviews Biophysics, 2014, Deng et al., RNA, 2016). I therefore do not see why a test for normality is implemented. If the authors have SHAPE data that is nearly Gaussian, I would like to see it appended as Supp. Material.

Response: We did not intend to suggest that our SHAPE data are nearly Gaussian. We have applied the standard pathway of statistical analysis here.

Comment 13: Results, subsection “Additional Analysis of Normalized SHAPE-CE Data”: To facilitate differential reactivity analysis between two samples and/or to assess reproducibility across experiments, the authors implemented a Mann-Whitney U test, aka Wilcoxon rank-sum test. Note that a very similar test, namely, a Wilcoxon signed-rank test, was recently used for differential reactivity analysis in Choudhary et al., Genome Biology, 2019, albeit applied to similarity scores, not reactivities. The Wilcoxon rank-sum test makes an independence assumption, whereas the signed-rank version of the test relaxes this assumption. Can the authors justify their assumption that the compared samples are independent, especially if these are replicates of the same experiment? Additionally, note that the proposed test does not account for biological variation between samples, since one needs >= 2 replicates from each condition/experiment to assess biological variation per condition. What this means is that if one performs the experiment on two biological replicates (especially relevant to in vivo samples), the test might indicate that results are not reproducible because it picks up the biological variation in the RNA’s structure and its reactivity to the SHAPE reagent. I think it is important to alert users to the limitation of the proposed test and to emphasize what it is capable of detecting. This limitation is inherent to any differential analysis test that compares between reactivity profiles without pre-assessing biological variability between samples in a condition.

Response: RNAthor incorporates statistical tests used by many labs (including Weeks lab) in which CE experiments are performed. These experiments result in obtaining independent samples, therefore the choice of non-parametric Mann-Whitney test (if the user selected 2 experiments) and Kruskal-Wallis rank-sum test (for 3-5 experiments). Let us underline these tests are not our choice. We implemented solutions already applied successfully in many laboratories.

Comment 14: As with the automated QC routine, I am missing a demonstration of the utility and the validity of the proposed statistical test. This could be done via benchmarking against existing methods (see, for example, Choudhary et al., Genome Biology, 2019 for a benchmark of recent methods). Potential users need to see some concrete evidence that the proposed test has statistical power. Alternatively, at the least, the authors should show examples of reproducible and irreproducible traces and their corresponding test scores as well as examples of how the test performs on biological replicates vs technical replicates. A comparison with existing reproducibility (QC) measures could also be helpful.

Response: Unfortunately, we do not understand this comment. Why should we validate and show statistical power of statistical tests which exist from years? This is the duty of the authors of the tests not of its users. See also REsponse to the previous comment.

Comment 15: It is clear that the authors have gained tremendous experience analyzing CE-based structure probing data, and that the proposed platform is the result of years of empirical hands-on experience. This is invaluable, especially since there aren’t many labs with that level of expertise. For this reason, this work has the potential to lead to real advances in data analysis. In particular, the authors say “RNAthor was extensively tested on ShapeFinder output files (IPF) from published and unpublished SHAPE-CE experiments performed in our laboratory. It was also tested on IPF files from hydroxyl radicals and DMS probing experiments resolved by capillary electrophoresis.” I think what’s really missing in this manuscript, other than an up-to-date review of existing work, is a demonstration of improved performance on many real datasets from multiple probes, multiple conditions, multiple labs, and multiple RNAs. Such demonstration should also include comparisons to existing alternatives, and there are several ones other than ShapeFinder and QuShape.

Response: We thank the Referee for appreciating our expertise in analyzing CE-based data. We would very much like to test our tool on many data sets from other laboratories. Unfortunately, it is not possible because other labs do not publish raw data from capillary electrophoresis. We have put a lot of effort into searching for such data and contacting other laboratories to obtain data for testing but we were unsuccessful. We also searched the literature, since some authors claimed to provide their raw data in supplementary materials. Unfortunately, it turned out that even in cases, where the authors of the publication attached supplementary files described as raw data, the files contained normalized data. Therefore, it is not possible to demonstrate the performance of RNAthor for multiple probes from external labs. We could only do it for data obtained in our laboratory, and we already did this before our paper submission. However, we agree that a demonstration of the tool's performance is important. Thus, we substantially revised the Results section and we described exemplary comparative experiments performed with RNAthor. We compared the results obtained by RNAthor with the results received from manual analysis done by the expert, for exemplary SHAPE and DMS data.

---

## [Decision Letter · Decision Letter 1]

27 Aug 2020

PONE-D-20-09557R1

RNAthor – fast, accurate normalization, visualization and statistical analysis of RNA probing data resolved by capillary electrophoresis

PLOS ONE

Dear Dr. Szachniuk,

Thank you for submitting your manuscript to PLOS ONE. After careful consideration, we feel that it has merit but does not fully meet PLOS ONE’s publication criteria as it currently stands. Therefore, we invite you to submit a revised version of the manuscript that addresses the points raised during the review process.

The revised manuscript should address all the critical points raised by all reviewers.

We look forward to receiving your revised manuscript.

Kind regards,

Danny Barash

Academic Editor

PLOS ONE

Reviewers' comments:

Reviewer's Responses to Questions

**Comments to the Author**

1. If the authors have adequately addressed your comments raised in a previous round of review and you feel that this manuscript is now acceptable for publication, you may indicate that here to bypass the “Comments to the Author” section, enter your conflict of interest statement in the “Confidential to Editor” section, and submit your "Accept" recommendation.

Reviewer #1: (No Response)

Reviewer #2: (No Response)

2. Is the manuscript technically sound, and do the data support the conclusions?

Reviewer #1: Yes

Reviewer #2: Partly

3. Has the statistical analysis been performed appropriately and rigorously? 

Reviewer #1: Yes

Reviewer #2: N/A

4. Have the authors made all data underlying the findings in their manuscript fully available?

Reviewer #1: No

Reviewer #2: Yes

5. Is the manuscript presented in an intelligible fashion and written in standard English?

Reviewer #1: Yes

Reviewer #2: Yes

6. Review Comments to the Author

Reviewer #1: I thank the authors for considering my suggestions. The revised manuscript presents a better case for the utility of RNAthor. The authors have explained why it is not necessary to have a standalone version of RNAthor at this time, added relevant references, as well as addressed my other minor comments.

A good number of experimental biologists are indeed unskilled at even basic data analysis tasks. Such biologists could benefit from a web server where they can upload outputs of ShapeFinder or QuShape and save normalized data, relevant figures and results from statistical tests. Hence, as a software tool RNAthor seems to have utility.

The primary contribution of the manuscript is a web server that combines existing functionalities from other sources. The method for automatic identification of unreliable data is an implementation of the rules that they have utilized previously (e.g., in their articles in Nature Communications, RNA Biology). The statistical tests that they perform have been used by other researchers active in the field. So overall, there appears to be no novelty with regards to the methods.

In my previous review, my major concern was that the method to identify “unreliable probing data” has not been validated. In revision, the authors have shown that the results from RNAthor are comparable to those from their manual analysis. This serves as evidence that there are no bugs in the software and that the rules followed by the human analyst have been faithfully implemented. The differences that they observed can be explained by subjectivity of manual analysis. However, objective validation of the method is still lacking.

The authors claim that comparison of RNAthor with manual analysis is the only possible validation. I believe that at the very least, the authors could test a range of cutoffs to identify and exclude unreliable data. The results from different cutoffs could be objectively compared by examining the accuracy in reproducing well-studied RNA structures, or other biological results that are widely believed in the field to be true. In my assessment, stating that an implemented algorithm is based on experience is not enough to claim novelty of the method. To make such a claim, the authors must demonstrate that out of a set of other plausible and reasonable methods, the method implemented in the web server is the one that performs the best. The authors can take a look at the following paper from the Laedarach lab as an example. In this article, Woods and Laedarach automated some of the rules used by humans for manual analysis. They tested a range of algorithms to identify the one that performs the best.

Woods, C. T., & Laederach, A. (2017). Classification of RNA structure change by ‘gazing’at experimental data. Bioinformatics, 33(11), 1647-1655.

In summary,

1. I find the article acceptable if it is classified as reporting a software tool. I’d recommend that the authors tone down their claim of novelty with regards to “an algorithm for the automatic identification of unreliable probing data” as this claim requires objective validations that are not there. I understand that the authors may have added this claim in response to my earlier comment about lack of novelty. My comment was based on assessing the manuscript as a methods research article, which may not be the intention of the authors. So it is best to remove this claim. Also, the authors should make the raw data used for figures 3-4 available as supplementary data, or provide links if they are available online.

2. To be accepted as a methods research article, major revisions are still needed to demonstrate that the method is indeed optimal and applicable for general use. Lack of gold standards to objectively evaluate methods for analysis of RNA structure-probing data is a challenge faced by all methods researchers active in the field. However, the community has also found acceptable ways for such evaluation. If the authors did indeed mean to publish RNAthor as a methods research article, I hope that the authors will borrow ideas from other manuscripts and consider validating their method.

Reviewer #2: The authors added several useful features to their web server, such as options to perform and visualize data-directed secondary structure prediction

and to analyze DMS data. Hopefully, this would make the proposed tool more appealing to potential users, although overall, the main novelty in this work

remains a fairly simple routine for automated detection of unreliable data points. A comparison between manual and automated data processing helps

demonstrate that the automated detection is reliable/judicious, although I would expect to see more examples if I were a potential user, or at least manual

analysis by additional experts (not the makers of the tool). Regarding the optional statistical tests, I think there are more statistically-sound and

powerful differential reactivity analysis methods out there, and I don’t feel what’s offered by this tool is very powerful.

Finally, please note that the graphics is of poor quality and should be improved.

7. PLOS authors have the option to publish the peer review history of their article (what does this mean?). If published, this will include your full peer review and any attached files.

Reviewer #1: No

Reviewer #2: No

---

## [Author Response · Author response to Decision Letter 1]

31 Aug 2020

Response to the comments of Referee #1:

COMMENT:

I thank the authors for considering my suggestions. The revised manuscript presents a better case for the utility of RNAthor. The authors have explained why it is not necessary to have a standalone version of RNAthor at this time, added relevant references, as well as addressed my other minor comments. A good number of experimental biologists are indeed unskilled at even basic data analysis tasks. Such biologists could benefit from a web server where they can upload outputs of ShapeFinder or QuShape and save normalized data, relevant figures and results from statistical tests. Hence, as a software tool RNAthor seems to have utility.

The primary contribution of the manuscript is a web server that combines existing functionalities from other sources. The method for automatic identification of unreliable data is an implementation of the rules that they have utilized previously (e.g., in their articles in Nature Communications, RNA Biology). The statistical tests that they perform have been used by other researchers active in the field. So overall, there appears to be no novelty with regards to the methods.

In my previous review, my major concern was that the method to identify “unreliable probing data” has not been validated. In revision, the authors have shown that the results from RNAthor are comparable to those from their manual analysis. This serves as evidence that there are no bugs in the software and that the rules followed by the human analyst have been faithfully implemented. The differences that they observed can be explained by subjectivity of manual analysis. However, objective validation of the method is still lacking.

The authors claim that comparison of RNAthor with manual analysis is the only possible validation. I believe that at the very least, the authors could test a range of cutoffs to identify and exclude unreliable data. The results from different cutoffs could be objectively compared by examining the accuracy in reproducing well-studied RNA structures, or other biological results that are widely believed in the field to be true. In my assessment, stating that an implemented algorithm is based on experience is not enough to claim novelty of the method. To make such a claim, the authors must demonstrate that out of a set of other plausible and reasonable methods, the method implemented in the web server is the one that performs the best. The authors can take a look at the following paper from the Laedarach lab as an example. In this article, Woods and Laedarach automated some of the rules used by humans for manual analysis. They tested a range of algorithms to identify the one that performs the best.

Woods, C. T., & Laederach, A. (2017). Classification of RNA structure change by ‘gazing’at experimental data. Bioinformatics, 33(11), 1647-1655.

RESPONSE: 

We agree that presenting the results of extensive and sophisticated computational tests of the new program significantly increases its credibility. However, it is difficult to perform many tests in the absence of raw data that other laboratories do not provide. The idea of testing a range of cutoffs is very interesting. It is quite easy to implement for a single-criterion problem, but our procedure is a multi-criteria decision problem. RNAthor has several input parameters (which the user can turn on/off) and internal parameters (bgArea, areaDifference, effectiveMaximum) - they depend on one another and all of them impact the results. We do not see the possibility to test these dependencies in a reasonable time.

COMMENT:

In summary,

1. I find the article acceptable if it is classified as reporting a software tool. I’d recommend that the authors tone down their claim of novelty with regards to “an algorithm for the automatic identification of unreliable probing data” as this claim requires objective validations that are not there. I understand that the authors may have added this claim in response to my earlier comment about lack of novelty. My comment was based on assessing the manuscript as a methods research article, which may not be the intention of the authors. So it is best to remove this claim. Also, the authors should make the raw data used for figures 3-4 available as supplementary data, or provide links if they are available online.

RESPONSE: 

From the very beginning, we intended to present a new software tool, not a new method. We agree that the paper should be classified as reporting a software tool. As recommended, we removed the claim of novelty regarding the unreliable data identification.

We also prepared the file with supplementary materials that includes raw data used for Figures 3-4. These data are also available as example data in the RNAthor web application.

COMMENT:

2. To be accepted as a methods research article, major revisions are still needed to demonstrate that the method is indeed optimal and applicable for general use. Lack of gold standards to objectively evaluate methods for analysis of RNA structure-probing data is a challenge faced by all methods researchers active in the field. However, the community has also found acceptable ways for such evaluation. If the authors did indeed mean to publish RNAthor as a methods research article, I hope that the authors will borrow ideas from other manuscripts and consider validating their method. 

RESPONSE:

We agree that our paper is not a methods research article. I suppose the misunderstanding was caused by the fact that sometimes we use the terms algorithm, method, procedure, and software tool interchangeably. To clear this issue, we have introduced several modifications in the manuscript.

 

Response to the comments of Referee #2:

COMMENT:

The authors added several useful features to their web server, such as options to perform and visualize data-directed secondary structure prediction and to analyze DMS data. Hopefully, this would make the proposed tool more appealing to potential users, although overall, the main novelty in this work remains a fairly simple routine for automated detection of unreliable data points. A comparison between manual and automated data processing helps demonstrate that the automated detection is reliable/judicious, although I would expect to see more examples if I were a potential user, or at least manual analysis by additional experts (not the makers of the tool). Regarding the optional statistical tests, I think there are more statistically-sound and powerful differential reactivity analysis methods out there, and I don’t feel what’s offered by this tool is very powerful.

Finally, please note that the graphics is of poor quality and should be improved.

RESPONSE: 

Referring to the suggestion to include more example data, we would like to note that although we currently have one example for SHAPE and one example for DMS data on the RNAthor webserver, these are complex examples. Each of them consists of three raw data files obtained for a relatively large RNA. The user can download the example files and rework them - creating new examples (e.g. load only two data files instead of three, cut out the data for a selected particle, shorten the particle sequence). It seems to us that there is a large field here to test our program using the files we provided. We would be happy to give more examples if there are scientists from other labs who provide us with their data. Unfortunately, we have not managed to find raw CE data publicly available, which we could use. We still have sets of data from our laboratory, which we can add as running examples to the RNAthor webserver as soon as we publish the structures that have been determined based on these data.

As for the statistical tests, we agree that various statistical methods could be implemented apart from the ones we provided. So far, we included basic tests, which - to our knowledge - are used commonly in many labs (including Week’s lab and our lab). But we are always open to suggestions of our users and we often add new options to our web servers if the users request them and explain why the new options are necessary. In the future, we will gladly enrich RNAthor with new features, including more powerful statistical tests.

Poor quality of the figures in the manuscript is probably due to very high compression on the PLOS server side. We can also see it in the pdf version generated by the submission server. All drawings in the original version meet PLOS requirements (tiff format, resolution 400dpi) and are of sufficient quality.

---

## [Editor Report · Decision Letter 2]

3 Sep 2020

RNAthor – fast, accurate normalization, visualization and statistical analysis of RNA probing data resolved by capillary electrophoresis

PONE-D-20-09557R2

Dear Dr. Szachniuk,

We’re pleased to inform you that your manuscript has been judged scientifically suitable for publication and will be formally accepted for publication once it meets all outstanding technical requirements.

Kind regards,

Danny Barash

Academic Editor

PLOS ONE
---

## [Editor Report · Acceptance letter]

22 Sep 2020

PONE-D-20-09557R2 

RNAthor – fast, accurate normalization, visualization and statistical analysis of RNA probing data resolved by capillary electrophoresis 

Dear Dr. Szachniuk:

I'm pleased to inform you that your manuscript has been deemed suitable for publication in PLOS ONE. Congratulations! Your manuscript is now with our production department. 

Kind regards, 

on behalf of

Dr. Danny Barash 

Academic Editor

PLOS ONE